# Phenotypical Differences between *Leishmania* (*Leishmania*) *amazonensis* PH8 and LV79 Strains May Impact Survival in Mammal Host and in Phlebotomine Sand Flies

**DOI:** 10.3390/pathogens12020173

**Published:** 2023-01-22

**Authors:** Fabia Tomie Tano, Erich Loza Telleria, Felipe Dutra Rêgo, Felipe Soares Coelho, Eloiza de Rezende, Rodrigo Pedro Soares, Yara Maria Traub-Cseko, Beatriz Simonsen Stolf

**Affiliations:** 1Department of Parasitology, Institute of Biomedical Sciences, University of São Paulo, São Paulo 05508-000, SP, Brazil; 2Laboratório de Biologia Molecular de Parasitas e Vetores, Instituto Oswaldo Cruz, Fiocruz, Rio de Janeiro 21040-360, RJ, Brazil; 3Department of Parasitology, Faculty of Science, Charles University, 12844 Prague, Czech Republic; 4Biotechnology Applied to Pathogens (BAP), Instituto René Rachou, Fundação Oswaldo Cruz (Fiocruz), Belo Horizonte 30000-000, MG, Brazil

**Keywords:** *L.* (*L.*) *amazonensis*, virulence, infection, complement, LPG, sand fly

## Abstract

We previously showed that *L.* (*Leishmania*) *amazonensis* promastigotes and amastigotes of the PH8 strain generated larger lesions in mice than LV79, and that lesion-derived amastigotes from the two strains differ in their proteomes. We recently reported that PH8 promastigotes are more phagocytized by macrophages. Promastigotes’ membrane-enriched proteomes showed several differences, and samples of each strain clustered based on proteomes. In this paper, we show phenotypic differences between PH8 and LV79 promastigotes that may explain the higher virulence of PH8. We compared in vitro macrophage infections by day 4 (early) and day 6 (late stationary phase) cultures, resistance to complement, and LPG characteristics. PH8 promastigotes showed a higher infectivity and were more resistant to murine complement. LPG was different between the strains, which may influence the interaction with macrophages and survival to complement. We compared the infection of the permissive vector *Lutzomyia longipalpis*. PH8 was more abundant in the vector’s gut 72 h after feeding, which is a moment where blood digestion is finished and the parasites are exposed to the gut environment. Our results indicate that PH8 promastigotes are more infective, more resistant to complement, and infect the permissive vector more efficiently. These data suggest that PH8 is probably better adapted to the sand fly and more prone to survive in the vertebrate host.

## 1. Introduction

Leishmaniasis is a group of diseases caused by *Leishmania* spp., with a broad clinical spectrum and epidemiological variety. The several clinical manifestations of the disease can be grouped into two main clinical forms: cutaneous (CL) and visceral (VL) leishmaniasis [1]. According to WHO reports, 98 countries were considered endemic for leishmaniasis in 2020, of which, 89 were for CL and 79 for VL [2]. The clinical manifestations and the disease severity are determined mainly by the host immune response and the *Leishmania* species and strains [3,4,5,6]. Around 20 species of *Leishmania* may cause human diseases, grouped into the subgenera *Leishmania*, *Viannia*, and *Mundinia* [7,8,9].

*Leishmania* promastigotes from *Leishmania* and *Viannia* subgenera are transmitted to mammals by female sand flies of the genus *Phlebotomus* or *Lutzomyia* during their blood meals [10]. Once in the vertebrate host, promastigotes are taken by phagocytes and differentiate into amastigotes inside macrophages [3,11]. To survive inside the hosts, the parasite relies on several molecules involved in “invasion and evasion”, termed virulence factors [3,12]. The most studied factors in *Leishmania* are glycoprotein 63 (GP63) and lipophosphoglycan (LPG), abundant molecules of the promastigote coat that contribute to resistance to complement lysis, parasite phagocytosis, and intracellular survival, among others [13,14,15,16,17,18,19].

GP63 is the most abundant protein on the promastigote surface [17], also found in other cellular compartments and shed in exosomes or by cleavage of its GPI anchor [20]. In the extracellular matrix, GP63 cleaves collagen and fibronectin, favoring promastigote dispersion [21]. GP63 cleaves C3b into C3bi, interrupting the complement cascade and allowing parasite internalization by CR3 [13]. Inside the macrophage, the enzyme cleaves several substrates, manipulating host cell signaling [20]. LPG, a membrane-anchored glycoconjugate, is a multivirulence factor involved in several functions both in the vertebrate and invertebrate host [22,23,24]. LPG affects promastigote binding to the Old World vector midgut, a crucial process for avoiding parasite loss with the digested blood meal [25], although no such clear function was seen in New World vectors [26].

Recent studies showed LPG polymorphisms in *L.* (*L.*) *amazonensis* strains (PH8 and Josefa). These variations were not determinant during the interaction with murine macrophages and permissive vectors *Lutzomyia migonei* and *Lutzomyia longipalpis*. However, the strains were not able to develop in restrictive *Phlebotomus papatasi* [27,28]. Furthermore, LPG polymorphisms (type I–III) in several *Leishmania (L.) infantum* strains did not affect the interaction with *L. longipalpis* [29]. Finally, LPG galactosylation levels affected the *Leishmania (L.) major* interaction with both permissive and restrictive vectors [30]. Altogether, these data indicate that the role of LPG polymorphisms is driven not only by *Leishmania* species but also by strains.

*L.* (*L.*) *amazonensis* is frequently associated to CL in Brazil [31], and symptomatic infections lead to a spectrum of clinical manifestations, including localized cutaneous leishmaniasis (LCL), anergic diffuse (ADCL), muco-cutaneous leishmaniasis (MCL), and even visceral leishmaniasis (CVL). In the case of ADCL, a lack of cell-mediated immunity and therapeutic failure hinders patient treatment [32,33,34].

In this study, we compared PH8 and LV79 strains of *L.* (*L.*) *amazonensis* isolated from sand fly and rodent, respectively. We have previously shown that PH8 promastigotes and amastigotes are more virulent to mice than LV79 [5]. We have recently reported that promastigotes from PH8 bind more and are more efficiently phagocytized by macrophages than those of LV79 [35]. The proteomes of membrane-enriched fractions from promastigotes of these strains have several differences, and although LV79 displayed higher amounts of GP63, PH8 showed increased GP63 activity [35].

Here, we compared some functional and biochemical aspects of the two strains that may contribute to the higher virulence of PH8. We compared PH8 and LV79 promastigotes in terms of infectivity, susceptibility to human and mice serum, LPG composition, and levels of infection in sandflies. The two strains display differences in all of these traits, some of which may contribute to PH8 virulent phenotype.

## 2. Material and Methods

### 2.1. Leishmania Promastigotes Culture

*L.* (*L.*) *amazonensis* and *L.* (*L.*) *infantum* promastigotes were cultured in 199 medium (Sigma Aldrich, Burlington, MA, USA) supplemented with 0.005% hemin, 40 mM HEPES pH 7.4, 100 µM adenine, 4 mM sodium bicarbonate, 20 μg/mL gentamicin, and 10% FBS. Promastigotes of LV79 (MPRO/BR/72/M1841, obtained from the rodent *Proechimys* sp. from Pará State, Brazil) and PH8 (IFLA/BR/67/PH8, isolated from the sand fly *Lutzomyia flaviscutellata* from Pará State, Brazil) strains were obtained through cultivation of amastigotes derived from BALB/c mice lesions in medium 199 at 24 °C. Parasites were subcultured weekly to an initial density of 2 × 10^6^ parasites/mL until the eighth passage. For that, BALB/c mice infected with LV79 and with PH8 were kept in our animal facility. Amastigotes were collected from the footpad lesions when promastigotes’ cultures reached 6–7 passages.

All experiments employed parasites at fourth day of culture (early stationary phase); macrophage infections employed day 4 and day 6 cultures.

*L.* (*L.*) *infantum* Ba262 strain (MCAN/BR/1989/BA262) was used as control for LPG assays.

### 2.2. Macrophage Infection

Bone-marrow-derived macrophages (BMDM) were differentiated from BALB/c mice femur as described in Tano et al., 2022 [35]. For infections, macrophages were harvested and counted, and 4 × 10^5^ were plated over coverslips in 24-well plates and incubated overnight at 37 °C under 5% CO_2_. Cells were infected at an MOI of 10:1 with day 4 and 6 promastigotes at 34 °C. After four hours, the medium was changed, and the plates were incubated for another 20 h at 34 °C and 5% CO_2_. Cells were fixed with methanol and stained with the Instant Prov Kit dye set (Newprov, Pinhais, Brazil). Assays were performed in technical triplicates and 100 cells were counted from each cover slip. The percentage of infected macrophages and the number of amastigotes per macrophage were calculated.

### 2.3. SDS-PAGE and Western Blot

Gels and *Western blot* were performed as previously described [35].

### 2.4. LPG Characterization

LPGs from procyclic parasites were extracted in solvent E (H_2_O/ethanol/diethylether/pyridine/NH4OH; 15:15:5:1:0.017) and purified using phenyl-sepharose. To confirm purification, 5 µg of LPG was submitted to immunoblotting using mAb CA7AE (1:1000), specifically to the phosphorylated Gal-(β1,4)-Man disaccharide repeats epitope [28]. To ascertain repeat units structures, LPGs were depolymerized and subjected to fluorophore-assisted carbohydrate electrophoresis (FACE) [36]. Briefly, repeat units were fluorescently labeled with 0.05 N ANTS (8-aminonaphthalene-1,3,6-trisulfate) and 1 M cyanoborohydride (37 °C, 16 h). Labeled sugars were subjected to FACE and the gel was visualized under UV light. Oligoglucose ladders (G1–G7) were used as standards for oligosaccharides gels [36].

### 2.5. Immunofluorescence

Promastigotes from the two strains were centrifuged, washed with PBS, and fixed with 4% paraformaldehyde for 30 min. Parasites were washed in PBS, transferred to glass slides, and air dried. Slides were incubated for 30 min in 50 mM ammonium chloride and for 10 min with TBS containing 1% BSA and 0,1% Triton, and then blocked for 30 min with PBS containing 1% de BSA at 37 °C. Cells were then incubated with anti-LPG (Thermo Fisher Scientific, Waltham, MA, USA) 1:1000 overnight at 4 °C. Slides were washed with PBS and incubated with DAPI (10 mg/mL) and anti-mouse Alexa fluor 488 (Thermo Fisher Scientific, USA) for 1 h at 4 °C. Slides were washed 10 times in PBS, 3 times in water, and fixed with ProLong^®^ (Life Technologies, Carlsbad, CA, USA).

### 2.6. Complement Lysis Assay

Human and BALB/c mice blood samples were incubated at room temperature for 15 min and centrifuged at 1500× *g* for 10 min. Sera were transferred to new tubes and stored at −80 °C. A total of 50 µL of PBS with 5 mM glucose containing human or murine sera from 0 to 40% was added to each well of a 96-well plate. Then, 1 × 10^7^ promastigotes in 50 µL of PBS containing 5 mM glucose and 2 mM of MgCl_2_ were added. The plate was incubated at 37 °C for 30 min and reaction was stopped with 20 µL of 60 mM EDTA. Parasite viability/metabolic activity was determined by MTT. Briefly, 20 µL of MTT (5 mg/mL) was added to each well and the plate was incubated at 24 °C for 50 min. Then, 100 µL of 10% SDS was added, and readings were taken at 595 nm.

### 2.7. Insects

Female *L. longipalpis* originally collected in Jacobina, Bahia, were kept in an insectary at 26 °C ± 2 °C, with a relative humidity of 80%. Insects were fed on 70% sucrose ad libitum and females were blood fed on anesthetized hamsters once a week (authorized by the Fiocruz Ethics Committee for the Use of Animals (CEUA) registration L-036/2019).

#### 2.7.1. *Lutzomyia longipalpis* Artificial Infection

For artificial infection of sandflies, defibrinated rabbit blood was used. Red blood cells were centrifuged and washed in sterile PBS 3 times and serum was inactivated at 56 °C for 30 to 45 min. The parasites were collected on early log phase culture, resuspended in the inactivated serum, and later mixed with the red blood cells. Artificial infections were performed with blood containing 10^7^ parasites/mL of blood using a Hemotek artificial feeder.

#### 2.7.2. RNA Purification and cDNA Synthesis of *Leishmania*-Infected Sandflies

*L. longipalpis* females were collected in groups of 10 insects at times after artificial infection (24, 72, and 168 h) and stored in Trizol (Life Technologies, USA). RNA extractions were performed according to the manufacturer’s instructions. After DNase treatment for removing possible DNA contaminants, cDNA synthesis was performed according to the reverse transcriptase manufacturer’s protocol (SuperScript Reverse Transcriptase-Invitrogen, Waltham, MA, USA).

#### 2.7.3. Quantification of Parasites in *L. longipalpis*

The presence of parasites was evaluated by quantitative PCR, using primers for the constitutive *Leishmania* actin gene. The expression of this gene was normalized as a function of an RP49 ribosomal protein gene from *L. longipalpis.*

## 3. Results

### 3.1. Macrophage Infection by PH8 and LV79 Promastigotes

We have already shown that PH8 early stationary phase promastigotes bind more and are more efficiently phagocytized by murine macrophages after 5 min of contact compared to LV79 [35]. This short initial interaction between the host cell and *Leishmania* promastigotes is very important and may favor either intracellular survival or parasite elimination depending on the molecules involved [37].

To analyze if the higher phagocytosis of PH8 promastigotes by macrophages leads to a higher infection rate after a longer period, we analyzed macrophage infection and parasite loads after 24 h. Infections were performed with promastigotes collected at day 4 and 6 of culture, corresponding to early stationary and late stationary phases, as reported [35]. Figure 1 shows infection levels from a representative experiment, indicating that PH8 infects a higher percentage of macrophages and leads to higher parasite loads. A higher infectivity was observed for both PH8 early and late stationary phase promastigotes.

Based on these results, all subsequent experiments were performed with early stationary phase promastigotes to avoid the high number of dead parasites sometimes observed at day 6.

A higher infectivity may be the result of different parasite molecules, such as membrane proteins and LPG.

### 3.2. PH8 Promastigotes Are More Resistant to Complement Lysis

Resistance to complement is usually acquired during metacyclogenesis, and several molecules have been shown to be involved in resistance [13,22,23,38]. Susceptibility to lysis was tested in PH8 and LV79 day 4 promastigotes and is shown in Figure 2.

Our results show that promastigotes from the two strains are more susceptible to lysis by human than by mouse complement. In addition, promastigotes of PH8 and LV79 were quite similar in terms of susceptibility to human serum. In fact, subtle differences (lower than 5%) in the numbers of viable parasites were observed only after incubations with 10% and 20% of serum (Figure 2a). On the other hand, PH8 and LV79 differ in terms of susceptibility to BALB/c mouse serum (Figure 2b). Indeed, PH8 was significantly more resistant to all concentrations tested.

Differences in LPG influence the infectivity in macrophages and affect the complement susceptibility, so we compared the LPG abundance and composition of PH8 and LV79.

### 3.3. LPG Composition Differs between PH8 and LV79 Strains

The *Western blot* of total extracts from promastigotes labeled with anti-LPG CA7AE antibody showed a higher labeling of LV79 extracts (Figure 3b). The immunofluorescence of fixed promastigotes also indicated a more intense labeling of LV79 (Figure 3c) by the CA7AE antibody [28]. These different labeling features between the strains suggest that the LPG expression varies between them.

To confirm if CA7AE recognition in both strains could be a result of masking the Gal-Man-P epitope by other membrane molecules, purified LPGs were subjected to a *Western blot*. Similarly, the strains were differentially recognized by the mAb CA7AE. The LPG from the PH8 strain was poorly labeled by this mAb, whereas those of LV79 and Ba262 (positive control) were intensely labeled (Figure 4).

To qualitatively assess the repeat unit composition of the above-mentioned LPGs, these glycoconjugates were depolymerized and subjected to FACE. Consistent with previous observations, the repeat units of the PH8 strain exhibited a lower disaccharide content, represented by Gal-Man (G_2_ position). This result explains the lower/absence recognition by CA7AE. Accordingly, in LV79 and Ba262 strains, the disaccharide content was higher (Figure 5).

LPG polymorphisms in *L.* (*L.*) *amazonensis* have already been reported in several strains from different clinical manifestations/hosts. In this species, LPGs were categorized according to the absence/presence of side chains in three types (I–III) [39]. Herein, FACE analysis revealed the absence of side chains in PH8 LPG (type I), different to what has previously been observed [28,39]. On the other hand, the repeat units of the LV79 strain showed one to two side chains, consisting of a type III LPG.

### 3.4. Phlebotomine Levels of Infection Differs between PH8 and LV79 Strains

LPG is not only associated to complement resistance and the binding of promastigotes to macrophage and macrophage inactivation, but may also be involved in phlebotomine sand fly infection. In addition, sand flies show immune responses to *Leishmania* [40], and the exposure of different virulence factors might impact the success of vector infection by the parasite. Since PH8 and LV79 strains differ in terms of membrane proteins [35] and in LPG biochemical composition, we analyzed promastigote persistence in sand fly at different times after artificial infection.

Permissive vectors such as *L. longipalpis* and *L. migonei* can sustain the infection of several *Leishmania* species, including *L.* (*L.*) *amazonensis* [12,19]. We thus infected *L. longipalpis* with PH8 and LV79 promastigotes and analyzed *Leishmania* abundance at different time points.

Data shown in Figure 6 indicate that PH8 parasites are more abundant in the vector’s gut than LV79 at 72 h after feeding. This adaptation may be related to differences in the parasite LPG and/or membrane protein composition or different immune responses of the insect to the parasite infection.

## 4. Discussion

The results presented here indicate that PH8 and LV79 promastigotes, already shown to differ in terms of proteomes, differ in other traits that influence the infection establishment in the vertebrate and invertebrate hosts.

In vitro infections of macrophages for 24 h were higher for PH8, in agreement with recently published data on phagocytosis [35].

Among the proteins that we previously identified as more abundant in PH8 [35] is the putative ABC transporter (*ATP-binding cassette*) member 1 from subfamily G (LABCG1), a protein located in the membrane and associated with virulence. Indeed, the deletion of LABCG1-2 in *L.* (*L.*) *major* reduced the promastigotes’ infectivity in vitro and lesion development in vivo [41], increased the parasite susceptibility to human complement lysis, and affected the LPG composition [41]. In this context, we wondered if PH8 and LV79 had differences in their resistance to complement and LPG composition. Promastigotes from both strains are more susceptible to lysis by human complement than by mouse complement. These findings agree with the lower levels of complement proteins and the lower lytic activity already reported for several mice lineages compared to other mammals [42]. Indeed, a lower efficacy of mouse serum was already registered against *L.* (*L.*) *major* metacyclic promastigotes [16]. Concerning lysis by mouse complement, PH8 was significantly more resistant than LV79. The proportion of metacyclics in promastigote’s culture affects complement resistance and infectivity. Unfortunately, the comparison of metacyclic proportions between *L.* (*L.*) *amazonensis* strains is not a simple task. Curiously, we have recently shown that morphometric and flow cytometry analyses indicated a higher proportion of metacyclics in LV79 day four cultures [35]. The expression of molecular markers commonly used for identifying metacyclics was similar between the two strains [35]. On the other hand, PH8 promastigotes display a higher abundance of LABCG1, possibly related to complement resistance in *L.* (*L.*) *major*. They also display higher GP63 proteolytic activity than LV79, despite having lower levels of this protein [35]. It is possible that the higher GP63 activity and the increased abundance of LABCG1 observed in PH8 contribute to its augmented resistance to complement, compensating for the eventual lower proportion of metacyclics observed in this strain. Since morphological analyses are more qualitative than quantitative and since markers showed no differences between the strains, we opted to use the same numbers of stationary phase promastigotes in the complement resistance and in the in vitro infection experiments.

Membrane proteins and LPG are known to affect promastigotes’ phagocytosis and resistance to complement. We compared the LPGs of PH8 and LV79 and observed a higher labeling of LV79 LPG with CA7AE, an antibody that recognizes repeat carbohydrate units of LPG. FACE analysis indicated that repeat units of PH8 LPG exhibited a lower disaccharide content, which may explain the lower reactivity against CA7AE. Curiously, a previous report comparing *L.* (*L.*) *amazonensis* PH8 and Josefa strains showed that the PH8 LPG displays galactose and mannose, as all LPGs, and a high content of glucose as sidechains, mainly as disaccharides [28]. The difference in the LPG composition of PH8 between this work and the previous report may be the result of our current growing conditions, and/or of the long duration of the culture of this strain in two independent labs. This is the first description of remarkable instraspecies polymorphisms in *L.* (*L.*) *amazonensis* strains.

We do not know the impact of these LPG discrepancies on the adhesion to macrophage, phagocytosis, and infection. Several reports showed that LPGs from several *Leishmania* species and strains induce different macrophage responses [28,43,44,45,46], and that LPGs with complex structures are usually associated to pro-inflammatory responses and nitric oxide (NO) production [29,45,46,47]. It is possible that LPG complexity affects other processes besides macrophage activation, and that the absence of side chains and a lower number of repetitive units could augment the access to other parasite membrane molecules.

In some *Leishmania*–phlebotomine combinations, LPG is the determinant for adhesion to the vector’s midgut. We have previously demonstrated that intraspecies LPG variations did not affect the interaction of *L.* (*L.*) *amazonensis* strains PH8 and Josefa with the permissive vectors *L. migonei* and *L. longipalpis* [27,28]. Still, in view of the differences between PH8 and LV79 strains in terms of the LPG (reported here) and protein composition (reported in [35]), we decided to explore their ability to infect *L. longipalpis*. A higher abundance of *Leishmania* was observed for the PH8 strain at 72 h of infection, a crucial moment when the blood digestion is coming to an end and a considerable number of parasites are eliminated during defecation, therefore suggesting that the PH8 strain survived this bottleneck better. Despite this fact, at later stages of the infection, both strains were able to sustain the infection in the vector. Previous data reported that other *L.* (*L.*) *amazonensis* and *L.* (*L.*) *infantum* strains displaying LPG polymorphisms did not differ in sand fly infection [27,28,29]. On the other hand, it is possible that all of the differences between these two *L.* (*L.*) *amazonensis* strains pointed out in this and in previous publications caused a distinct activation of the sand fly immunity, leading to different levels of infection. We have previously shown irrefutably that *L. longipalpis* presents immune responses to *Leishmania* infections. In the case of sand flies infected by *L.* (*L.*) *infantum*, the suppression of the Jak-STAT [48] or activation of IMD pathways [49] affected the survival of the parasite in the vector, and TGF-beta also had a role in the success of the infection [50]. We already showed that exosomes secreted by parasites from logarithmic and stationary phase cultures, equivalent to the insect forms, carry a plethora of *Leishmania* virulence factors [46]. In addition, *Leishmania* exosomes were detected inside the gut of infected sand flies, and proteomic studies also identified virulence factor in these vesicles [47]. Indeed, there is a complex interplay between *Leishmania* and their insect vectors, and the different levels of *L. longipalpis* infection by the two *Leishmania* strains might be the result of the insect response to parasite molecules secreted or presented on its surface. We are presently investigating these possibilities.

We believe that this work complements our previous study on the PH8 and LV79 promastigotes protein composition, and we hope that it stimulates further studies on LPG and proteins that may contribute to differential resistance to complement and adhesion to the permissive vector’s midgut.

## Figures and Tables

**Figure 1 pathogens-12-00173-f001:**
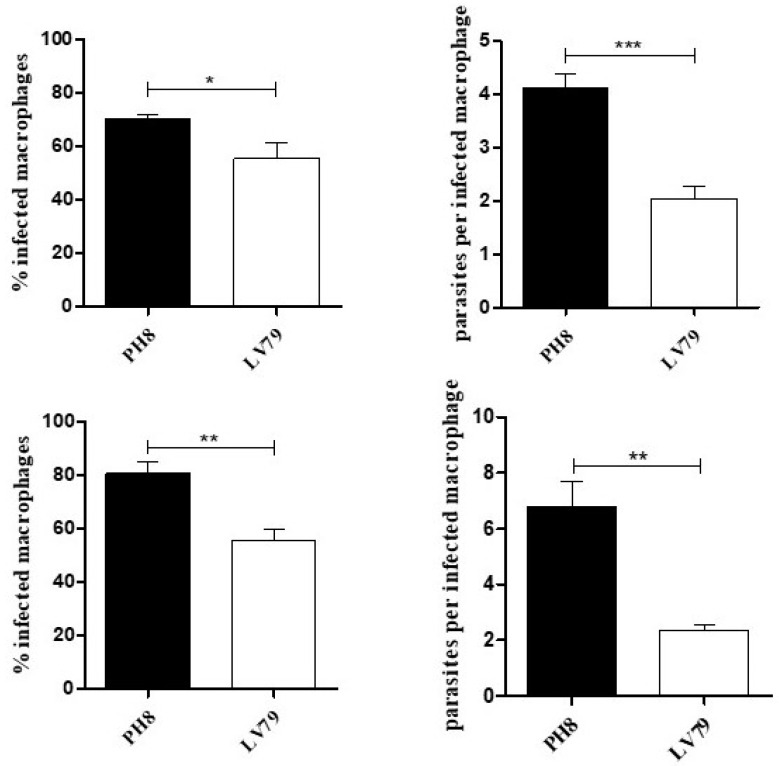
Infection of BMDMs with *L.* (*L.*) *amazonensis* PH8 and LV79 promastigotes from days 4 (upper) and 6 (lower) of culture. Graphs show percentage of infected macrophages and parasites per infected macrophage. Means and standard deviations from technical triplicates. Statistical analysis was performed by Student’s *t* test, *: *p* < 0.05, **: *p* < 0.01, ***: *p* < 0.001. Data from two experiments.

**Figure 2 pathogens-12-00173-f002:**
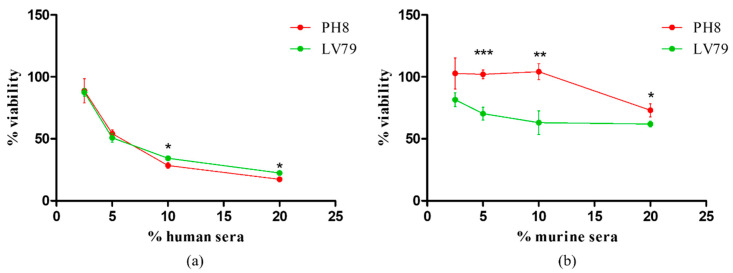
Resistance of *L.* (*L.*) *amazonensis* PH8 and LV79 promastigotes at day 4 of culture to human (**a**) and mouse (**b**) complement. Percentage of viable cells relative to control condition (no serum) after incubation for 30 min at 37 °C in the presence of 2.5%, 5%, 10%, or 20% of serum. Means and standard deviations from technical triplicates. Statistical analysis was performed by Student’s *t* test, *: *p* < 0.05, **: *p* < 0.01, ***: *p* < 0.001. Data from one experiment representative of three (**a**) and two (**b**) independent experiments with similar profiles.

**Figure 3 pathogens-12-00173-f003:**
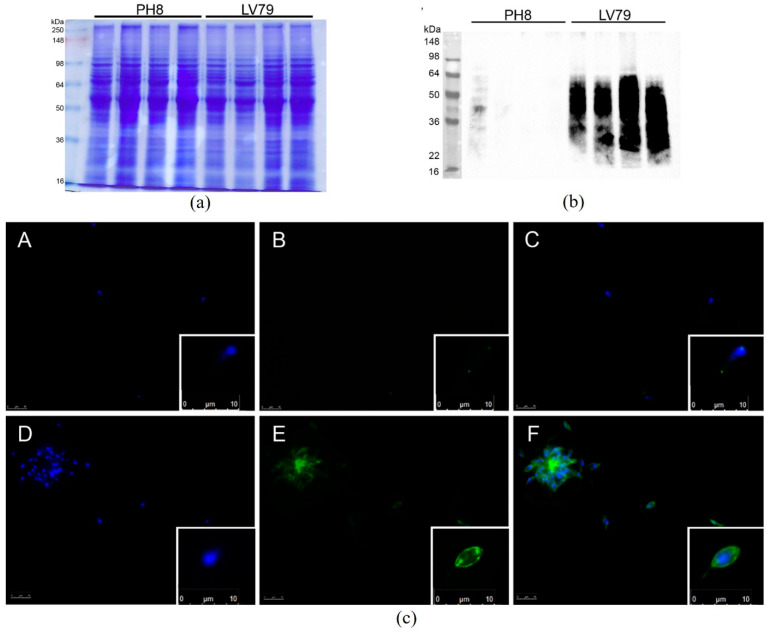
Analysis of LPG of *L.* (*L.*) *amazonensis* PH8 and LV79 promastigotes by *Western blot* and immunofluorescence employing CA7AE antibody. (**a**) SDS-PAGE showing PH8 and LV79 total extracts (4 independent samples for each strain). (**b**) *Western blot* for LPG using CA7AE antibody. (**c**) Immunofluorescence of promastigotes from PH8 ((**A**–**C**) upper images) and LV79 ((**D**–**F**) lower images) labeled with CA7AE antibody (4 independent samples for each strain). (**A**,**D**), labeling of nuclei with DAPI; (**B**,**E**), labeling of LPG (Alexa Fluor 488); (**C**,**F**), overlay. In the right border, detailed image of a representative promastigote.

**Figure 4 pathogens-12-00173-f004:**
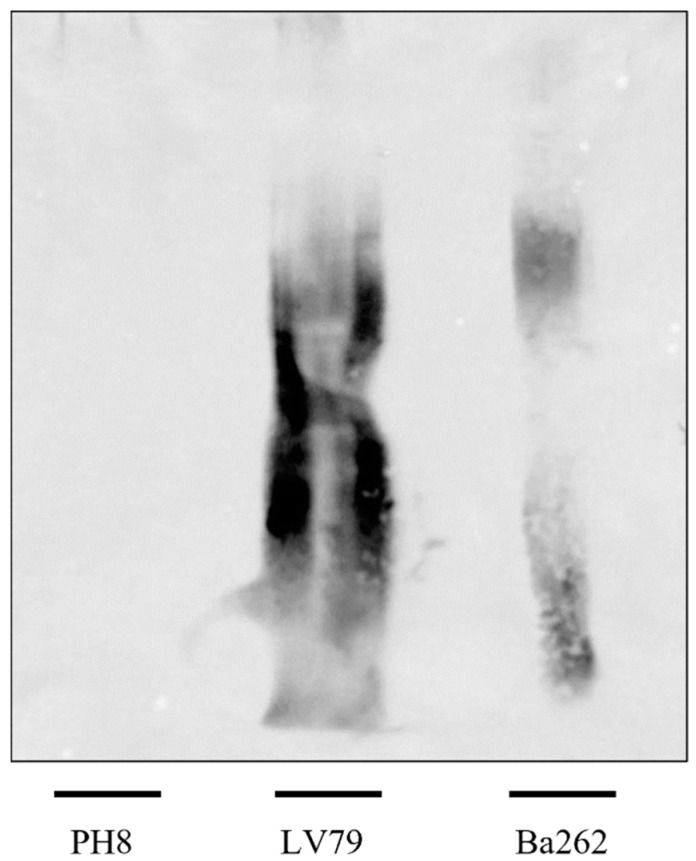
Immunoblotting of purified LPG (5 μg per lane) from *L.* (*L.*) *amazonensis* PH8 and LV79 strains and *L.* (*L.*) *infantum* Ba262 strain probed with mAb CA7AE. Lane 1, PH8 strain; Lane 2, LV79 strain; Lane 3, control strain represented by Ba262.

**Figure 5 pathogens-12-00173-f005:**
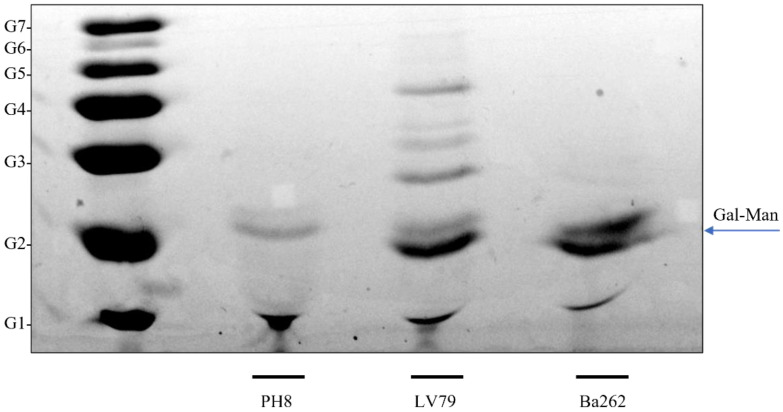
Fluorophore-assisted carbohydrate electrophoresis (FACE) of lipophosphoglycan (LPG) repeat units of *L.* (*L.*) *amazonensis* (PH8 and LV78 strains) and *L.* (*L.*) *infantum* Ba262 strain. Lane 1: oligoglucose ladder represented by 1–7 glucose residues (G_1_–G_7_); Lane 2, repeat units of PH8 strain; Lane 3, repeat units of LV79 strain; Lane 4, repeat units of Ba262 strain. Gal, galactose and Man, mannose.

**Figure 6 pathogens-12-00173-f006:**
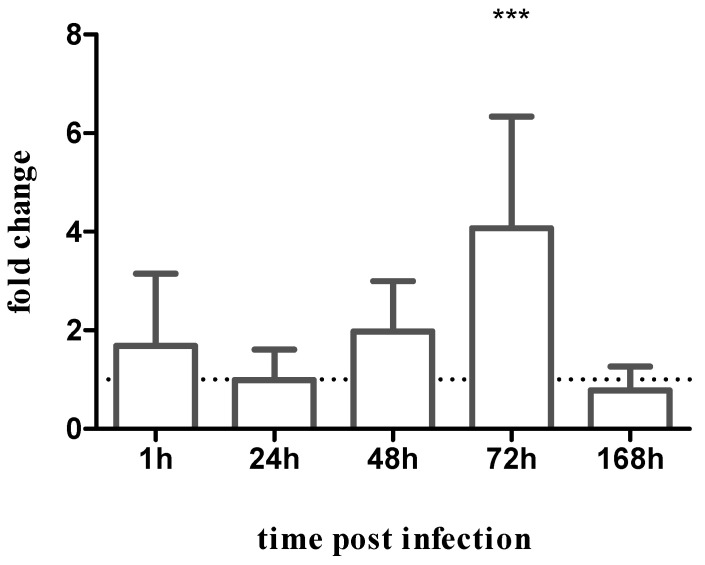
Quantification of PH8 compared to LV79 promastigotes in *L. longipalpis*: PH8 and LV79 *L.* (*L.*) *amazonensis* parasite presence was assessed in infected sand flies by qPCR. Y-axis represents *Leishmania* actin relative gene expression of PH8 strain normalized to the sand fly RP49 ribosomal protein gene and expressed as fold change compared to the LV79 strain (dotted line). X-axis represents infected sand fly samples collected at 1, 24, 48, 72, and 168 h post-infection. Bars represent the relative gene expression obtained from three independent experiments. Significant differences between PH8 and LV79 actin gene expression were tested using the two-way ANOVA method with Sidak’s multiple comparisons tests, ***: *p* < 0.001.

## Data Availability

Not applicable.

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
