# Peer review of "Phenotypical Differences between Leishmania (Leishmania) amazonensis PH8 and LV79 Strains May Impact Survival in Mammal Host and in Phlebotomine Sand Flies"

_pathogens, 2023, doi:10.3390/pathogens12020173_

Round 1

Reviewer 1 Report

The manuscript of Tano et al. Describes the phenotypic differences between two different L. amazonensis strains. Overall, the manuscript is well-written and the methodological methods described are suited to phenotypically characterize and compare parasites strains and species. However, I do have some experimental concerns about some of the methods used that do deserve some attention before publication.

Major comments:

Was promastigote growth of both strains compared during the in vitro infection experiments? If one of both strains would grow slower than the other (might even differ based on the type medium in which the culture is held) comparing day 4 and day 6 infections is pointless. Was the number of metacyclic parasites determined? If so, by which technique? If not, better to change the term into stationary phase parasites.

In l306 the authors do mention that the percentage of metacyclic parasites was higher for the LV79 strain than for the PH8 strain, which make it logical that when you than compare infectivity etc, that the LV79 has a huge advantage. It would be better and more logical to wok with equal number of metacyclic parasites of both strains. This should at least be considered or mentioned in the discussion.

Minor comments:

To improve readability of the manuscript the first sentence of the second-to last paragraph of the discussion (In this study, we compared…) could better be placed at l76 after the general intro on L. amazonensis.

The parasite strains that have been compared have already been isolated over 50 years ago. How have they been maintained in the lab since then? As virulence can alter over the course of time, this might also account for the differences observed between the strains at present day.

Have both strains been continuously cultivated in vivo in the lab to maintain virulence?

L111: start of new section – the title is missing its format and numbering

L147: what is the frequency of the feeding on hamsters? Were the flies also fed a sugar solution during blood feedings?

L151: what was the stage of parasites used for the infections? Were early log phase cultures used? How old were these?

L164: did the authors also evaluate the location and the anterior migration of the parasites inside the sandfly gut? Determining the number of parasites inside does not give additional information on the infectivity of the fly.

Can the authors explain the significant different response of both strains against the mouse serum compared to the human serum were the difference is much smaller?

Author Response

We would like to thank Reviewer 1 for the careful revision and for raising issues that led to an improvement of the manuscript. Please find our answers below. The changes are depicted in the revised version of the paper.

Major comments:

  1. Was promastigote growth of both strains compared during the in vitro infection experiments? If one of both strains would grow slower than the other (might even differ based on the type medium in which the culture is held) comparing day 4 and day 6 infections is pointless. Was the number of metacyclic parasites determined? If so, by which technique? If not, better to change the term into stationary phase parasites.

R: Thank you very much for raising such important issues.

We have compared growth curves of the two strains in 199 media, and they were quite similar. This data was published recently, in the paper describing proteomic differences between LV79 and PH8 promastigotes (Tano et al, 2022).

The comparison of metacyclic proportions between L. amazonensis strains is not a simple task. Purification of L. amazonensis metacyclics can be performed using antibody mAb 3A1 directed to procyclic LPG [59] or Ficoll gradients. Ficoll separation was developed for L. major [60], and although not based on LPG and feasible for purification of LPG deficient L. major, its efficiency is affected by LPG characteristics. In fact, gal rich LPGs such as the ones from L. major, L. tropica and L.amazonensis (Josefa strain, Nogueira et al., 2017) allow more efficient Ficoll enrichment of metacyclics, while LPGs with glucose side chains preclude efficient purification of metacyclics using Ficoll (Rodrigo Soares, personal communication). The LPG profile of the PH8 strain revealed presence of glucose as side chains, and for this reason we don´t consider the comparison of the proportion of metacyclics in PH8 and LV79 by Ficoll purification accurate.

In face of these technical caveats on metacyclic purification for comparison, in the previous paper by Tano already mentioned (Tano et al, 2022) we opted to estimate metacyclic proportions by two different morphological techniques and to compare the abundance of commonly used metacyclic markers in cultures of LV79 and PH8.

  1. In l306 the authors do mention that the percentage of metacyclic parasites was higher for the LV79 strain than for the PH8 strain, which make it logical that when you than compare infectivity etc, that the LV79 has a huge advantage. It would be better and more logical to work with equal number of metacyclic parasites of both strains. This should at least be considered or mentioned in the discussion.

R: The higher abundance of metacyclics in LV79 according to the morphological analyses was quite unexpected in face of the higher infectivity of PH8 cultures and the higher similarity of the growth curves of the cultures. The expression of molecular markers commonly used for identifying metacyclics was similar between the two strains. These data reinforce the complexity of classification of stages in L. amazonensis and defy the scientific community to invest in projects aimed to study L. amazonensis development in the vector and search for appropriate metacyclic markers for this species.

Since morphological analyses are more qualitative than quantitative and since markers showed no differences between the strains, we opted to use the same number of stationary phase promastigotes. We appreciate the comments of the reviewer, and we agree that this is not a simple issue. Three sentences mentioning these issues were included in the discussion.

The proportion of metacyclics in promastigote´s culture affects complement resistance and infectivity. Unfortunately, the comparison of metacyclic proportions between L. amazonensis strains is not a simple task. (…)

The expression of molecular markers commonly used for identifying metacyclics was similar between the two strains (…)

Since morphological analyses are more qualitative than quantitative and since markers showed no differences between the strains, we opted to use the same numbers of stationary phase promastigotes in the complement resistance and in the in vitro infection experiments

Minor comments:

  1. To improve readability of the manuscript the first sentence of the second-to last paragraph of the discussion (In this study, we compared…) could better be placed at l76 after the general intro on L. amazonensis.

R: We have changed the place of this sentence as suggested, and it really sounds better. Thank you very much.

  1. The parasite strains that have been compared have already been isolated over 50 years ago. How have they been maintained in the lab since then? As virulence can alter over the course of time, this might also account for the differences observed between the strains at present day.

Have both strains been continuously cultivated in vivo in the lab to maintain virulence?

R: This is a very important question. We always keep BALB/c mice infected with LV79 and with PH8 in our animal facility. Amastigotes are collected from the footpad lesions when promastigotes’ cultures reach 6-7 passages, so that we always employ promastigotes cultures until 8 passages.

We included this information in Material and Methods section. Thank you.

  1. L111: start of new section – the title is missing its format and numbering

R: Thank you. We have revised it.

  1. L147: what is the frequency of the feeding on hamsters? Were the flies also fed a sugar solution during blood feedings?

R: Insects were fed on 70 % sucrose ad libitum and females were blood fed on anesthetized hamsters once a week. We have added this information to item 2.7.

  1. L151: what was the stage of parasites used for the infections? Were early log phase cultures used? How old were these?

R: Infection of sand flies were done using promastigote parasites collected on early log phase culture, approximately 3 days old. We have now added this information to item 2.7.1.

  1. L164: did the authors also evaluate the location and the anterior migration of the parasites inside the sandfly gut? Determining the number of parasites inside does not give additional information on the infectivity of the fly.

R: Unfortunately, we did not evaluate the parasite localization in the sand fly gut. We do agree this could provide interesting data.

7 Can the authors explain the significant different response of both strains against the mouse serum compared to the human serum were the difference is much smaller?

R: We don´t have any hypothesis to explain the small but significant higher resistance to human complement observed in LV79 promastigotes, and the higher resistance of PH8 to mouse complement. Both strains were more susceptible to human than by mouse complement. As we mentioned in the paper, this can be due to the lower levels of complement proteins and the lower lytic activity reported for several mice lineages compared to other mammals. However, the lack of precise knowledge about the differences between human and mouse complement makes the interpretation of our results difficult.

Reviewer 2 Report

With the work entitled “Phenotypical Differences Between Leishmania (Leishmania) amazonensis PH8 and LV79 Strains May impact Survival In Mammal Host and In Phlebotomine Sand Flies” the authors described some phenotypic differences related to infection and survival of different strains of parasites in contact with the hosts .

These data are important for the biological knowledge of the species studied, considering some changes in its pathogenicity depending on the expression and activity of virulence factors, opening new perspectives.

I have a few suggestions to make in the text and figures:

1. Authors must check the spelling of the manuscript. It has several scientific names without being italicized, in addition to the lack of punctuation.

2. The BALB/c strain is classically susceptible to Leishmania amazonensis infection. Authors should also test at least the infection in the C57BL/6 mouse strain.

3. In figure 1 the authors should show the infection index as follows: 100 stained cells are inspected, and the percentage of infected macrophages is multiplied by the average number of amastigotes per macrophage.

4. The sentences on lines 111 and 112 are taken out of context within Material and Methods.

5. Considering that human serum was tested to measure susceptibility to the complement system, why were human lineage or primary macrophages not used to verify the comparative level of infection?

6. Regarding non-recognition by the anti-LPG used in the PH8 strain, how can the authors clarify in the manuscript that there are quantitative LPG differences between species using an antibody that does not recognize this molecule in PH8? This implies difference only in composition. This observation in the amount of LPG present is not clear in immunofluorescence.

7. How to know if the insects fed on the same amount of amastigotes? This can generate a distorted analysis referring to Figure 6.

8. Figure 6 could be presented in another graph. The way the authors explored the graph became confusing through the representation of the dotted line. For a better understanding, try to redo it by plotting the corresponding bars of the strains used on the same graph.

Author Response

We would like to thank Reviewer 2 for the careful revision of our manuscript. Please find our answers below.

  1. Authors must check the spelling of the manuscript. It has several scientific names without being italicized, in addition to the lack of punctuation.

R Thank you very much for the careful revision. We apologize for the mistakes.

  1. The BALB/c strain is classically susceptible to Leishmania amazonensisinfection. Authors should also test at least the infection in the C57BL/6 mouse strain.

R: We do agree that different results can be obtained in different mouse strains. We have already analyzed infections with LV79 and PH8 promastigotes in a previous study: de Rezende et al., 2017. Although lesions were much smaller in C57BL/6 mice, there was also a significant difference between LV79 and PH8 lesion sizes. Please find the results obtained in the figure attached.

  1. In figure 1 the authors should show the infection index as follows: 100 stained cells are inspected, and the percentage of infected macrophages is multiplied by the average number of amastigotes per macrophage.

R: When we don´t have extensive number of strains or time points we usually prefer to show both parameters (percentage of infection and parasite number) instead of the index. We believe they can give more precise information than their product (infection index).

  1. The sentences on lines 111 and 112 are taken out of context within Material and Methods.

R: Thank you. They were not numbered. We have revised that.

  1. Considering that human serum was tested to measure susceptibility to the complement system, why were human lineage or primary macrophages not used to verify the comparative level of infection?

R: We have used mouse and human complement and observed more pronounced differences between the two Leishmania strains when exposed to mouse complement. We chose to compare infection in mouse macrophages due to the higher differences observed for the complement of this species and to the availability of mouse infection model employed in previous works with LV79 and PH8. Besides, experiments with medullary and peritoneal murine macrophages are easy to perform and usually very reproducible.

  1. Regarding non-recognition by the anti-LPG used in the PH8 strain, how can the authors clarify in the manuscript that there are quantitative LPG differences between species using an antibody that does not recognize this molecule in PH8? This implies difference only in composition. This observation in the amount of LPG present is not clear in immunofluorescence.

R: We respectfully disagree with the reviewer since CA7AE antibody is the mAb developed to recognize the conserved Gal-Man disaccharides present in all LPGs (Tolson et al., 1989). PH8 shows low labeling in total extracts (Fig3) and in purified LPG (Fig4), indicating lower LPG content. Consistent with those observations, lower disaccharide content of PH8 LPG represented by a fainted band at G2 position was also shown by FACE analysis (Fig5). Altogether those data reinforced that the lack/lower recognition by CA7AE was due to LPG content.

  1. How to know if the insects fed on the same amount of amastigotes? This can generate a distorted analysis referring to Figure 6.

R: 107 culture promastigotes/mL of blood were given to the insects for both isolates through artificial infection as described in Methods. Our qPCR data of infected sand flies at 1h post infection shows no significant differences between strains, indicating that feeding was probably similar.

  1. Figure 6 could be presented in another graph. The way the authors explored the graph became confusing through the representation of the dotted line. For a better understanding, try to redo it by plotting the corresponding bars of the strains used on the same graph.

R: The graph in figure has a commonly used format for qPCR calculation using DDCt method where one group of samples is taken as the reference group, in our case sand flies infected by LV79 strain, while the second group, PH8, is considered the analyzed group. The difference between the two groups is expressed as fold change compared to the LV79, which has the proportion value equal to one (1). We believe that the originally submitted graph format will not create confusion to the readers, but if the reviewer find it better, we can provide an alternative graph format as shown in the attached file.

Round 2

Reviewer 1 Report

Given the complexity of some of the methodological approaches that have carefully and in detail been explained by the authors, I can now support publication of the manuscript.